EXOSC10 is a novel hepatocellular carcinoma prognostic biomarker: a comprehensive bioinformatics analysis and experiment verification

Meng Zhi-Yong 1 2
Fan Yu-Chun 3
Zhang Chao-Sheng 1
Zhang Lin-Li 1
Wu Tong 1
Nong Min-Yu 1
Wang Tian 1
Chen Chuang ch1ch2@163.com 4
Jiang Li-He jianglihe@ymun.edu.cn 1 3 5 6
1 School of Basic Medical Sciences, Youjiang Medical University for Nationalities , Nanning , China
2 First Clinical Medical College, Guangxi University of Traditional Chinese Medicine , Nanning , China
3 Medical College, Guangxi University , Nanning , China
4 Guangxi Medical University Cancer Hospital , Nanning , China
5 Key Laboratory of Minimally Invasive Techniques & Rapid Rehabilitation of Digestive System Tumor of Zhejiang Province,Taizhou , Zhejiang , China
6 Special Key Laboratory of Gene Detection & Therapy of Guizhou Province (Zunyi Medical University) , Guizhou , China
Zhang Xin
Electronic publication date: 2023 Sep 8
Publication date: 2023
Volume: 11
Electronic Location ID: e15860
Received 2023 Jan 13; Accepted 2023 Jul 17
Copyright: ©2023 Meng et al.
Copyright year: 2023
Copyright holder: Meng et al.
License: This is an open access article distributed under the terms of the Creative Commons Attribution License, which permits unrestricted use, distribution, reproduction and adaptation in any medium and for any purpose provided that it is properly attributed. For attribution, the original author(s), title, publication source (PeerJ) and either DOI or URL of the article must be cited.
License URL: https://creativecommons.org/licenses/by/4.0/

Keywords: Hepatocellular carcinoma, EXOSC10, Prognostic biomarker, Bioinformatics analysis, Experiment verification

Funding: Youjiang Medical College for Nationalities YY2021SK02 Foundation of Nanning Qingxiu District Key Research and Development Project 2020023 Open Project of Guangxi key Laboratory of Enhanced Recovery after Surgery for Gastrointestinal Cancer GXEKL202203 Open Project Program of Key Laboratory of Minimally Invasive Techniques & Rapid Rehabilitation of Digestive System Tumor of Zhejiang Province MIRRLAB 21SZDSYS13 Grant of Special Key Laboratory of Gene Detection & Therapy of Guizhou Province (Zunyi Medical University QJKH-KY [2017]007-002 National-level Project of University Students’ Innovation and Entrepreneurship in 2022 202210599005 202210599007 Guangxi Provincial-Level Project of University Students’ Innovation and Entrepreneurship in 2022 S202210599056 This study was supported by the Grant of research project on high-level talents of Youjiang Medical College for Nationalities (Grant No. YY2021SK02), the Foundation of Nanning Qingxiu District Key Research and Development Project (Grant No. 2020023), the Open Project of Guangxi key Laboratory of Enhanced Recovery after Surgery for Gastrointestinal Cancer (Grant No. GXEKL202203), the Open Project Program of Key Laboratory of Minimally Invasive Techniques & Rapid Rehabilitation of Digestive System Tumor of Zhejiang Province (Grant No. MIRRLAB 21SZDSYS13), the Grant of Special Key Laboratory of Gene Detection & Therapy of Guizhou Province (Zunyi Medical University) (No. QJKH-KY [2017]007-002), the Grant of National-level Project of University Students’ Innovation and Entrepreneurship in 2022 (Grant No. 202210599005; 202210599007), the Grant of Guangxi Provincial-Level Project of University Students’ Innovation and Entrepreneurship in 2022 (Grant No. S202210599056). The funders had no role in study design, data collection and analysis, decision to publish, or preparation of the manuscript.

==============================
Background

Hepatocellular carcinoma (HCC) is a common malignant tumor. There are few studies on EXOSC10 (exosome component 10) in HCC; however, the importance of EXOSC10 for HCC remains unclear.

Methods

In the study, the prognosis value of EXOSC10 and the immune correlation were explored by bioinformatics. The expression of EXOSC10 was verified by tissue samples from clinical patients and in vitro experiment (liver cancer cell lines HepG2, MHCC97H and Huh-7; normal human liver cell line LO2). Immunohistochemistry (IHC) was used to detect EXOSC10 protein expression in clinical tissue from HCC. Huh-7 cells with siEXOSC10 were constructed using lipofectamine 3000. Cell counting kit 8 (CCK-8) and colony formation were used to test cell proliferation. The wound healing and transwell were used to analyze the cell migration capacity. Mitochondrial membrane potential, Hoechst 33342 dye, and flow cytometer were used to detect the change in cell apoptosis, respectively. Differential expression genes (DEGs) analysis and gene set enrichment analysis (GSEA) were used to investigate the potential mechanism of EXOSC10 and were verified by western blotting.

Results

EXOSC10 was highly expressed in tissues from patients with HCC and was an independent prognostic factor for overall survival (OS) in HCC. Increased expression of EXOSC10 was significantly related to histological grade, T stage, and pathological stage. Multivariate analysis indicated that the high expression level of EXOSC10 was correlated with poor overall survival (OS) in HCC. GO and GSEA analysis showed enrichment of the cell cycle and p53-related signaling pathway. Immune analysis showed that EXOSC10 expression was a significant positive correlation with immune infiltration in HCC. In vitro experiments, cell proliferation and migration were inhibited by the elimination of EXOSC10. Furthermore, the elimination of EXOSC10 induced cell apoptosis, suppressed PARP, N-cadherin and Bcl-2 protein expression levels, while increasing Bax, p21, p53, p-p53, and E-cadherin protein expression levels.

Conclusions

EXOSC10 had a predictive value for the prognosis of HCC and may regulate the progression of HCC through the p53-related signaling pathway.

Introduction

Hepatocellular carcinoma (HCC) is a common malignant tumor of the digestive system throughout the world. There were 905,677 new cases of hepatocellular carcinoma, representing 4.7% of new cancer cases, with 830,180 deaths (8.3%) in 2020 (Sung et al., 2021; Zhou et al., 2020). As a common malignancy, many factors have been proven to be involved in its occurrence and development, such as chronic hepatitis C, chronic hepatitis B, and hereditary hemochromatosis (De Sanctis et al., 2020). Currently, treatment methods for patients with early liver cancer include local radio frequency ablation, partial hepatectomy, and liver transplantation (Liang et al., 2021; Rando-Segura et al., 2019; Xu et al., 2021). However, almost 70% of patients will relapse in five years after the operation. Despite advances in early detection and drug development, the prognosis for patients with HCC remains poor, which poses a challenge for clinical therapists (Oweira et al., 2017; Zhang & Zhang, 2019; Zhou et al., 2021). Thus, uncovering novel and reliable biomarkers is an urgent need to improve clinical outcomes and reduce the burden of cases.

The 3′–5′ exoribonuclease EXOSC10 (also called PM/Scl-100 in humans or Rrp6 in yeast and fly) is part of the multimeric nuclear RNA exosome and interacts with numerous proteins Kowalinski et al., 2016; Januszyk, Liu & Lima, 2011, which has broad clinical importance because it is the target of autoantibodies produced in patients suffering from polymyositis/scleroderma overlap syndrome (Jamin et al., 2017). Depletion of EXOSC10 can have an impact on exoribonucleolysis of nuclear RNA metabolism and transcriptional control and lead to increased levels of dilncRNA and DNA-RNA hybrids (Davidson et al., 2019; Domingo-Prim et al., 2019). Furthermore, the depletion of EXOSC10 in human cell lines can stabilize short poly (A) RNAs and increase the length of their poly (A) tails (Wu & Dean, 2020). Although, recent evidence showed that human EXOSC10 is an unfavorable prognostic marker for liver cancer (Uhlen et al., 2017). However, the role of EXOSC10 in HCC is unclear and the prognostic value and function of EXOSC10 in HCC still remain to be elucidated.

In this investigation, to elucidate the diagnosis and predict the prognostic value of EXOSC10 in HCC. We investigated the prognostic role of EXOSC10 in patients with HCC through bioinformatics analysis, clinical samples, and in vitro experiments. The results of the study will further strengthen our understanding of the biological mechanism of EXOSC10.

Materials and Methods

Sample information

The mRNA expression data and related clinical data of HCC were downloaded from The Cancer Genome Atlas (TCGA) database (https://www.cancer.gov/ccg/research/genome-sequencing/tcga/) (Cancer Genome Atlas Research Network et al., 2013), which included patient ID, age, survival time, survival status, tumor grade, stage, and stage of TNM. And the missing and uncertain clinical data were deleted. If there were multiple samples from the same patient, the average value of mRNA expression of the same gene was obtained in the samples, and the deadline was March, 2021. Data extraction and merging were performed using Perl (version 5.3.0). Finally, we obtained 374 HCC patient samples and 50 normal control samples from the TCGA database for a follow-up study.

The TCGA database is open to the public, and all data have been agreed to be used for analysis and have obtained moral recognition. This study is based on open source data, strictly abides by the release guidelines and access policies of the database, and is not bound by other ethics.

EXOSC10 mRNA expression analysis

The analysis of EXOSC10 mRNA expression was based on the TCGA database using the package R “Limma”, and the “ggplot2” package was used to draw a box plot for visualization.

The Shiny Methylation Analysis Resource Tool (SMART) database (http://www.bioinfo-zs.com/smartapp/) was used to investigate the distribution of methylation sites in EXOSC10 and pan-cancer analysis of EXOSC10 methylation (Li, Ge & Lu, 2019). The expression of EXOSC10 in pan-cancer was identified in the TIMER database (http://timer.cistrome.org) (Li et al., 2017).

Survival and clinicopathological analysis in HCC patients

The correlation between EXOSC10 expression and patient prognosis, such as overall survival (OS), and progression-free survival (PFS), was analyzed in HCC and shown by forest plots, nomogram, and Kaplan–Meier by R package “survival” and “RMS”. The multivariate or univariate Cox regression models were performed using R package “survival”. Logistic regression analysis was used to explore the correlation of EXOSC10 expression with clinical factors of HCC using the R package “Limma”.

GO annotation enrichment and KEGG pathway enrichment analysis

The samples in TCGA were divided into high and low EXOSC10 expression groups according to the median level of EXOSC10 mRNA. Differential expression genes (DEGs) between high and low EXOSC10 expression groups were identified using R package “Limma”, with absolute logarithmic fold change (FC) >1 and FDR < 0.05 were considered the cutoff criterion. Gene ontology (GO) and KEGG pathway analysis were performed using DEGs in high and low expression groups of EXOSC10 using the R package “clusterProfiler”. P < 0.05 was considered statistically significant for GO annotation enrichment analysis and KEGG pathway enrichment analysis.

GSEA (gene set enrichment analysis)

Potential EXOSC10-regulated pathways were analyzed by GSEA. We used GSEA software to investigate the tumor hallmarks between high and low EXOSC10 gene expression. P < 0.05 and a false discovery rate (FDR) < 0.25 were considered statistically significant.

Correlation analysis of EXOSC10 with immune cell infiltration

TIMER (https://cistrome.shinyapps.io/timer/) was used to analyze the correlation of EXOSC10 expression with six kinds of tumor-infiltrating immune cells (B cells, CD8+ T cells, CD4+ T cells, macrophages, neutrophils, and dendritic cells). Furthermore, we analyzed the association of the immune response of immune cells with EXOSC10 expression in HCC by CIBERSORT (Newman et al., 2015). P < 0.05 was considered statistically significant.

Furthermore, we applied the “Estimation of Stromal and Immune cells in Malignant Tumors using Expression data” (ESTIMATE) (Yoshihara et al., 2013) algorithm to assess the relationship between EXOSC10 expression and immune/stromal/estimate scores in each HCC samples.

Correlation analysis of EXOSC10 with immune checkpoints and TMB analysis

The correlation of EXOSC10 expression with immune checkpoints in HCC was evaluated using R package “Limma” and “corrplot” based on TCGA. Results with P < 0.001 were considered statistically significant. In addition, the correlation of EXOSC10 expression with tumor mutation burden (TMB) was calculated by R package “Limma”.

Immunohistochemistry (IHC)

We collected 20 pairs of HCC samples and adjacent non-tumor tissues from October 2018 to April 2019 in the Affiliated Hospital of Youjiang Medical College for Nationalities. Immunohistochemistry (IHC) was performed on the 20 samples to detect EXOSC10 protein expression by DAB staining.

All procedures were performed according to the Ethical Guidelines for Human Genome/Gene Research and approved by the Ethics Committee of Affiliated Hospital of Youjiang Medical College for Nationalities (2022090501). Furthermore, we received verbal informed consent from participants of our study.

Cell culture

The liver cancer cell lines (HepG2, MHCC97H, and Huh-7) and normal human liver cell line LO2were purchased from Cell Bank of Type Culture, Shanghai Institute of Biochemistry and Cell Biology, Chinese Academy of Sciences. Cells were cultured in DMEM, supplemented with 10% FBS (Fetal Bovine Serum) and 1% antibiotic/antimycotic solution. Cells were maintained at 37 °C in an atmosphere of 5% CO2.

RNA extraction and quantitative real-time PCR

Total RNA was extracted using AxyPrep Multisource Total RNA Miniprep Kit (Cat No: Scipu003342; Axygen, Suzhou, China), and for mRNA quantification, RNA was reverse transcribed to cDNA using the Revert Aid First Strand cDNA Synthesis Kit (Cat No: K1621; Thermo Fisher Scientific, Waltham, MA, USA) according to the manufacturer’s protocol. Quantitative real-time PCR was conducted using the FastStart Universal SYBR Green Master (Cat No: FSUSGMMRO; Roche, USA). The 2−ΔΔCt method was used for quantitative. Folding changes of target genes are standardized through internal control.

The following primers were used:

EXOSC10(forward: 5′-GGAACCGTAAGGCAGCAGAAT-3′; reverse: 5′-TCTCGAAACTTGAGCTGAGGT-3′);

GAPDH (forward: 5′-GGACCTGACCTGCCGTCTAG-3′; reverse: 5′-GTAGCCCAGGATGCCCTTGA-3′).

Western blotting

Western blot was carried out according to the manufacturer’s instructions. The RIPA buffer with 1% phenylmethanesulfonyl fluoride was used to lyse cells, and extracted proteins were separated by 10% SDS-PAGE. Furthermore, the proteins were transferred onto polyvinylidene fluoride (PVDF) membranes. The membranes were blocked with skim milk for 2 h at room temperature, incubated with primary antibodies overnight at 4 °C, and were washed by Tris–HCl solution + Tween-20 (TBST) three times for 10 min. Subsequently, they were incubated with appropriate secondary antibodies for 1 h at room temperature. Finally, the blots were detected with enhanced chemiluminescence (ECL Plus). The density of protein bands was assessed by Image J software, and the protein levels were normalized to GAPDH. The dilution ratio of the antibody used is as follows: EXOSC10 (Cat No. 16731-1-AP, 1:3000; Proteintech, Wuhan, China), p53 (Cat No. AF0879, 1:1000; Affinity, Jiangsu, China), p-p53 (Cat No. AF3073, 1:1000; Affinity, Jiangsu, China), p21 (Cat No. AF6290, 1:1000; Affinity, Jiangsu, China), Bax (Cat No. 50599-2-Ig, 1:1000; Proteintech, Wuhan, China), Bcl-2 (Cat No.68103-1-Ig, 1:1000; Proteintech, Wuhan, China), PARP (Cat No. AF7023, 1:1000; Affinity, Jiangsu, China), N-cadherin (Cat No.22018-1-AP, 1:5000; Proteintech, Wuhan, China), E-cadherin (Cat No.20874-1-AP, 1:5000; Proteintech, Wuhan, China), GAPDH (Cat No.60004-1-Ig, 1:20000; Proteintech, Wuhan, China).

Transfection

EXOSC10 siRNA and negative control (NC) siRNA were purchased from GenePharma (Shanghai, China). The siRNAs and plasmids were transfected into cells by lipofectamine 3000 (Cat No. L3000015; Invitrogen Life Technologies, Carlsbad, CA, USA).

For siRNA knockdown, Huh-7 cells were seeded in 6-well plates. After 24 h, siRNA transfections were performed using Opti-MEM and Lipofectamine3000 transfection reagent according to the manufacturer’s instructions, transfected for 48 h, then harvested for further assays. The transfection effect was detected by Western blotting and quantitative real-time PCR, respectively. The following primers were used: Negative Control (NC) (5′–3′, S′-UUCUCCGAACGUGUCACGUTT; AS′-ACGUGACACGUUCGGAGAATT); siEXOSC10-2201 (5′–3′, S′-GCAGCGAAGUUC GAUCCAUTT; AS′-AUGGAUCGAACUUCGCUGCTT).

Cell Counting Kit-8 (CCK-8) assay

The growth inhibition rate of Huh-7 cells was assessed using the CCK-8 assay (Cat No. CK04; Dojindo Laboratories, Japan) assay. Cells (5 × 103 cells per well) were seeded in 96-well plates with three replicate wells. After transfected for 24 h, 48 h, 72 h, and 96 h, a 10 µl volume of CCK-8 reagent was added to each well. Then measure the absorbance at 450 nm after incubation with the CCK-8 solution at 37 °C for 1 h. The inhibition rate (%) = (the OD values of NC − the OD values of siEXOSC10) / (the OD values of NC − the OD values of blank) ×100%.

Wound healing assay

Huh-7 cells were seeded in 6-well plates, after 48 h of transfection, cell confluence reached 90%. Then the 10 µl pipette tip was used to gently scratch the cell monolayer, washed twice with PBS, and removed the detached cells. The remaining cells were cultured in DMEM medium without FBS. The scratch areas were photographed at 0 and 24 h, respectively.

Migration assay

Twenty-four-well transwell plates were used to detect cell migration. Collect the cells and add them separately to the top compartments of the 24 well transwell plate. Meanwhile, add DMEM medium containing 10% FBS to the bottom of the transwell chamber. After 24 h, cells passing through the insert were stained with crystal violet and counted under a phase contrast microscope. Randomly select five fields and calculate the average number of cells inserted.

Colony formation assay

After 48 h of transfection, cells were collected and 1,000 cells were seeded in each well of a 6-well plate and cultured in DMEM medium containing 10% FBS for 2 weeks. The medium was changed after one week and then changed twice a week. After 2 weeks, cells were washed twice with phosphate buffer saline (PBS) and fixed in one mL of 4% paraformaldehyde fixation and methanol in each well for 20 min, respectively. After aspiration of methanol, the cells were washed twice with PBS again, stained with crystal violet staining solution for 20 min, photographed and counted.

Mitochondrial membrane potential measurement

The JC-1 kit (Cat No. C2006; Beyotime Biotechnology, Shanghai, China) was used to detect the membrane potential. NC or transfected cells were seeded in a 6-well plate and stained with JC-1 by incubation in culture solution for 20 min, and then each group of cells was observed with a fluorescence microscope.

Hoechst33342 accumulation assay

Hoechst 33342 dye stains (Cat No. C1025; Beyotime Biotechnology, Shanghai, China) were used in apoptosis detection. NC or transfected cells were seeded in a 6-well plate and stained with 1 ml of Hoechst 33342 dye by incubation in the culture solution for 30 min, then washed twice with PBS, observed each group of cells with a fluorescence microscope.

Flow cytometry

The Cell Meter™ FITC-Annexin V Binding Apoptosis Assay Kit (Cat No. 22839; AAT Bioquest, Newport Beach, CA, USA) was used to investigate the effect of EXOSC10 silencing on Huh-7 cell apoptosis. NC or transfected cells were cultured in a 6-well plate with an FBS-free medium for 48 h. Cells were resuspended in 200 µL assay buffer, 2 µL Annexin V-FITC, and 2 µL 100 × propidium iodide. The cells were also incubated in the dark at room temperature for 30 min. Finally, we analyze the cells with a flow cytometer.

Statistical analysis

Statistical analyzes were performed using R software v4.0.4 and SPSS 25.0. The one-way ANOVA or two-tailed unpaired Student’s t-test was performed to identify significant differences in the data. *P < 0.05, **P < 0.01, ***P < 0.001, N.S.: no significant difference (P  ≥ 0.05).

Results

Expression of EXOSC10 in HCC patients

The mRNA level of EXOSC10 in HCC patients was evaluated according to TCGA LIHC data sets. As shown in Fig. 1A, HCC samples tended to express higher levels of EXOSC10 mRNA than normal samples. Additionally, the level of EXOSC10 mRNA was also significantly increased in 50 HCC samples compared to their adjacent liver tissues (Fig. 1B).

Figure 1 The clinical features of EXOSC10 in HCC patients.

(A) EXOSC10 is more highly expressed in HCC samples compared with normal samples. (B) EXOSC10 is significantly increased in 50 HCC samples compared with their corresponding adjacent liver tissues. (C) The relationship between EXOSC10 expression and age. (D) The relationship between EXOSC10 expression and gender. (E) The relationship between EXOSC10 expression and grade. (F) The relationship between EXOSC10 expression and distant metastases. (G) The relationship between EXOSC10 expression and lymph node metastasis. (H) The relationship between EXOSC10 expression and clinical stage. (I) The relationship between EXOSC10 expression and tumor stage.

We evaluated EXOSC10 expression data by all patient characteristics derived from TCGA. In histological grade, we found that the expression of EXOSC10 in G3 was significantly increased in G1 (P = 0:0066) and G2 (P = 0:0065). The expression of EXOSC10 in stage I decreased significantly compared to that in stage II (P < 0.001) and stage III (P <  0.001). The expression of EXOSC10 in stage IV decreased compared to that in stage II (P = 0.033) and stage III (P = 0.018). In the T stage, the expression of EXOSC10 in T1 was significantly decreased than that in T2 (P < 0.001) and T3 (P < 0.001). However, EXOSC10 expression did not show any correlation with the M and N stages, respectively. These results suggested that the expression level of EXOSC10 was related to histological grade, T stage, and pathological stage (Figs. 1C–1I).

The correlation between EXOSC10 expression and prognosis in patients with HCC

We analyzed the correlation between EXOSC10 expression and the prognosis of patients with HCC. According to the Kaplan–Meier survival curve, HCC patients with high expression of EXOSC10 had poor overall survival (OS) and progression-free survival (PFS) (Figs. 2A, 2B). We established a time-dependent survival ROC curve based on the expression of EXOSC10 in the sample to predict 1-, 3-, and 5-year survival rates (AUC = 0.665, 0.620, and 0.565, respectively) (Fig. 2C). The results showed that the use of EXOSC10 expression to predict the prognosis of patients with HCC was reasonably accurate.

Figure 2 Expression of EXOSC10 was correlated with prognosis of HCC patients.

The association of EXOSC10 with (A) overall survival and (B) progression free survival in HCC. (C) ROC curves of EXOSC10 for predicting 1/3/5-year survival. Univariate (D) and multivariate (E) Cox regression analysis identified the association of EXOSC10 with the clinical factors. (F) A nomogram for predicting HCC prognosis by EXOSC10 expression. (G) Calibration plot of the nomogram for predicting the probability of OS at 1, 3, and 5 years.

Furthermore, univariate Cox regression analysis showed that EXOSC10 expression significantly affects the prognosis of patients with HCC. Multivariate Cox regression analysis showed that EXOSC10 can be independent of prognostic factors of clinical characteristics in patients with HCC patients (Figs. 2D, 2E). To predict the OS of patients with HCC, we established a nomogram based on the expression of EXOSC10 and patient characteristics in the TCGA dataset (Fig. 2F). The calibration plot for the probability of 1-, 3- and 5-year OS showed good agreement between the observed OS and nomogram-predicted OS (Fig. 2G). The above studies collectively demonstrate that EXOSC10 expression is significantly correlated with the prognosis of HCC patients.

Potential pathways regulated by EXOSC10 in HCC

HCC patients were divided into high and low EXOSC10 expression groups according to the median expression level of EXOSC10. To explore the potential pathway-regulated EXOSC10, we identified 2,724 significant DEGs between high and low expression groups, of which 41 genes were down-regulated and 2,683 genes were up-regulated (Text S1). The 50 most significant genes are shown in Fig. S1. Go enrichment analysis revealed that these DEGs were primarily enriched in biological processes such as nuclear division, organelle fission, and chromosome segregation; cellular components such as condensed chromosomes, synaptic membrane, microtubules; molecular functions such as channel activity, motor activity of microtubules and passive transmembrane transporter activity. The most enriched KEGG pathways included cell cycle, neuroactive ligand–receptor interaction, and ECM receptor interaction (Fig. 3A). While GSEA analysis indicated that several tumor hallmarks were enriched in the high EXOSC10 expression group, such as E2F target, MTORC1 signaling pathway, PI3K-AKT-MTOR signaling pathway, p53 signaling pathway, TGF beta signaling pathway, and WNT beta-catenin signaling pathway (Fig. 3B). The results may give some insight into the cellular biological effects related to EXOSC10.

Figure 3 GO enrichment analysis based on DEGs and GSEA enrichment analysis revealed the tumor hallmarks associated with EXOSC10 in HCC.

(A) GO annotations and KEGG pathways of DEGs in HCC. (B) Gene set enrichment analysis (GSEA) indicating that tumor hallmarks were enriched in high EXOSC10 expression group.

The correlation of EXOSC10 expression and tumor immune infiltration

The relationship between EXOSC10 expression and immune cell infiltration adjusted by purity (B cells, CD8+ T cells, CD4+ T cells, macrophages, neutrophils, and dendritic cells) was investigated using TIMER. As shown in Fig. 4A, EXOSC10 has a significant positive correlation with tumor purity (r = 0.092, P = 8.91e−02), B cells (r = 0.263, P = 7.515e−07), CD8+ T cells (r = 0.275, P = 2.26e−07), CD4+ T cells (r = 0.34, P = 9.33e−11), macrophages (r = 0.434, P = 4.00e−17), neutrophils (r = 0.451, P = 1.08e−18), and dendritic cells (r = 0 .421, P = 4.61e−16) in HCC. Next, we performed a correlation analysis between EXOSC10 and the tumor microenvironment, immune/stromal/estimate scores in HCC. Compared to the low EXOSC10 expression group, the high EXOSC10 expression group had a significant decrease in the stromal score and the estimate score (Fig. 4B). However, the potential correlation of EXOSC10 expression and tumor immune infiltration needs to be further tested experimentally. In addition, we compared the fraction of immunocytes with the high or low expression groups of EXOSC10. Unfortunately, only B cell memory with high expression of EXOSC10 was significantly higher than those with low expression of EXOSC10 (Fig. 4C). Finally, we evaluated the correlation between EXOSC10 and immune cells (Figs. 4D–4F). And we found that EXOSC10 had a significant positive correlation with B cell memory (r = 0.28, P = 0.042) and a negative correlation with B cell naiveness (r = −0.3, P = 0.031).

Figure 4 The association of EXOSC10 with immune infiltration in HCC.

(A) EXOSC10 expression is significantly associated with tumor purity and infiltrating of immune cells. (B) The relationship between EXOSC10 expression and immune/stromal/estimate scores in HCC. (C) Effect of differential EXOSC10 expression on immune cells infiltration. (D) Correlation between expression of EXOSC10 and immune cells infiltration. (E) Correlation between expression of EXOSC10 and B cells memory. (F) Correlation between expression of EXOSC10 and B cells naive. Note: ∗P < 0.05, ∗∗P < 0.01 and ∗∗∗P < 0.001.

Validation of EXOSC10 expression levels in HCC patients and liver cancer cell lines

Furthermore, we measured the expression of EXOSC10 mRNA and protein in three liver cancer cell lines (HepG2, MHCC97H and Huh-7) and the normal liver cell line LO2 by quantitative real-time PCR and Western blotting, respectively. These results confirmed that EXOSC10 expression levels were significantly higher in liver cancer cell lines than in control LO2cells (P < 0.05), consistent with the results of bioinformatic analysis (Figs. 5A–5B). These results suggested that up-regulation of EXOSC10 may be closely associated with the occurrence and development of HCC.

Figure 5 EXOSC10 expression in liver cancer.

(A) EXOSC10 protein expression in human liver cancer cell lines. (B) EXOSC10 expression in human liver cancer cell lines. (C) The protein expression of EXOSC10 in HCC by IHC. ∗P < 0:05, ∗∗P < 0:01 and ∗∗∗P < 0:001.

We also found that the expression of the EXOSC10 protein was up-regulated in HCC tissue compared to normal liver tissue by analyzing 20 pairs of HCC samples and adjacent non-tumor tissues (Fig. 5C). As shown in Fig. 5C, the positive area of EXOSC10 was higher in HCC than in normal liver tissue, and the expression of the EXOSC10 protein was located primarily in the cytosol and nuclear lumen.

Silencing EXOSC10 inhibited HCC cell proliferation and migration

To explore the function of EXOSC10 in liver cancer cells, we chose the Huh-7 liver cancer cell line to study. We transfected Huh-7 cells with EXOSC10 siRNA, resulting in a significant decrease in the level of expression of EXOSC10 mRNA and protein, suggesting effective elimination of EXOSC10 (Fig. 6A).

Figure 6 Silencing EXOSC10 inhibited HCC cell growth and migration.

(A) EXOSC10 expression determined by qRT-PCR and western blot assay, respectively. (B–C) Cell viability assessed by CCK8 assay and colony formation assay, respectively. (D–E) Cell migration determined by transwell assay and wound healing assay. ∗:P < 0:05; ∗∗:P < 0:01; ∗∗∗:P < 0:001.

Then we evaluated the function of EXOSC10 in the growth and proliferation of HCC cells. As shown in Fig. 6B, the knockdown of EXOSC10 inhibited the proliferation of HCC cells. Interestingly, silencing of EXOSC10 decreased the colony formation of HCC cells (Fig. 6C). Furthermore, we investigated whether EXOSC10 affects the migration ability of HCC cells. Wound healing and transwell migration assays demonstrated that silencing the expression of EXOSC10 decreased the migration of Huh-7 cells (Figs. 6D–6E).

Silencing EXOSC10 reduced mitochondrial membrane potential and promoted cell apoptosis in Huh-7 cells

Next, we investigate whether EXOSC10 affects cell apoptosis in Huh-7 cells. First, we explore the detection of cell apoptosis using Hoechst 33342 dye. The results showed that apoptosis in the siEXOSC10 group was significantly higher than in the NC group (Fig. 7A). We found that the mitochondrial membrane potential in the NC group was significantly higher than in the siEXOSC10 group (Fig. 7B). Similarly, we confirmed that the apoptosis rate in the NC group was significantly lower than in the siEXOSC10 group by flow cytometer (Fig. 7C).

Figure 7 Silencing EXOSC10 promoted cell apoptosis in Huh-7 cells.

(A) Silencing EXOSC10 promoted cell apoptosis by hoechst 33342 dye. (B) Silencing EXOSC10 reduced mitochondrial membrane potential. (C) Silencing EXOSC10 promoted cell apoptosis by flow cytometer. ∗:P < 0:05; ∗∗:P < 0:01; ∗∗∗:P < 0:001.

EXOSC10 regulated the p53 signaling pathway in Huh-7 cells

We found that PARP, N-cadherin, and Bcl-2 protein levels in the NC group were significantly higher than those of the siEXOSC10 group; protein levels of Bax, p21, p53, p-p53, and E-cadherin were significantly lower than those of the siEXOSC10 group (Fig. 8).

Figure 8 EXOSC10 regulated the p53 signaling pathway in Huh-7 cells.

The protein levels of PARP, N-cadherin, and Bcl-2 in NC group were significantly higher than those in siEXOSC10 group; the protein levels of Bax, p21, p53, p-p53, and E-cadherin were significantly lower than those in siEXOSC10 group. ∗:P < 0:05; ∗∗:P < 0:01; ∗∗∗:P < 0:001.

Discussion

Globally, HCC is the second leading cause of cancer-related death, with about 841,000 newly diagnosed cases and 782,000 deaths each year (Bray et al., 2018). As a malignant tumor with a poor prognosis, it is still of great significance to investigate the pathogenesis and prognostic factors of HCC (Yang & Heimbach, 2020). Previous studies have shown that mutations in EXOSC genes are associated with a variety of distinct diseases (de Amorim et al., 2020). It is reported that the expression of EXOSC10 might influence human development and disease progression (Stuparević et al., 2021). However, investigations on the expression, regulation, and prognostic significance of EXOSC10 in HCC are still lacking. In our research, the prognostic value and biological function of EXOSC10 were investigated using comprehensive bioinformatic methods and verified by experiments.

Based on the TCGA database, the higher levels of EXOSC10 mRNA in HCC patients and the prognostic value of EXOSC10 in HCC patients were evaluated. Furthermore, we also found that the expression of the EXOSC10 protein was up-regulated in HCC tissue compared to normal liver tissue by IHC. Poor OS and PFS were observed in HCC patients with high expressed EXOSC10. To investigate potential pathways regulated by EXOSC10 in HCC, patients with HCC were divided into high and low EXOSC10 expression groups according to the median expression level of EXOSC10. A total of 2,724 significant DEGs were identified between high and low expression groups, such as LAMP5, SEZ6L and FOXD1. Previous studies had shown that LAMP5 could serve as a prognostic signature to predict survival in patients with gastric cancer, and its high expression levels could accurately estimate KMT2A-r in acute leukemias (Wang et al., 2017; Lopes et al., 2022). Loss of normal function of SEZ6L could accelerate the progression of lung cancer and pancreatic cancer (Gorlov et al., 2007; Chen et al., 2022). FOXD1 is a member of the evolutionarily conserved Forkhead box (FOX) family of genes (Herman, Todeschini & Veitia, 2021). Numerous studies had shown that FOXD1 was known to promote the malignant processes of various types of cancers, such as pancreatic cancer, gastric cancer, and lung cancer (Cai et al., 2022; Wu et al., 2021; Li et al., 2019). Finally, based on the identified DEGs, functional analyses suggested that various pathways including cell cycle, ECM receptor interaction, and p53 signaling pathway may mediate the role of EXOSC10 in HCC.

DNA methylation is an epigenetic mechanism that is essential for regulating gene transcription, and may play a critical role in cancer development and progression (Mahmoud & Ali, 2019). The methylation of cytosines in DNA is a prominent modification associated with gene expression regulation. It is widely accepted that DNA methylation results in transcriptional repression by interfering with the binding of transcription activating factors or recruiting transcriptional repressors that reduce chromatin accessibility, eventually resulting in gene silencing (Ando et al., 2019). In this study, EXOSC10 was found to have a significantly lower methylation level in HCC than in normal tissues according to the SMART database (Fig. S2). Therefore, to some extent, we can conclude that it is the lower level of EXOSC10 DNA methylation that can result in higher levels of EXOSC10 mRNA and protein expression in HCC compared to normal tissue.

In recent years, increasing evidence showed that immunotherapy for HCC is important (Zongyi & Xiaowu, 2020; Shang et al., 2021). In this study, we explore the relationship between EXOSC10 expression and immune cell infiltration. The results showed that the expression of EXOSC10 was significantly correlated with immune cell infiltration in HCC, including B cells, CD8+ T cells, CD4+ T cells, macrophages, neutrophils and dendritic cells. And data from the ESTIMATE method showed that EXOSC10 was positively correlated with the stromal score and estimate scores. These results imply that EXOSC10 plays an important role in immune regulation.

Higher expression of immune checkpoint molecules usually benefits more from immune checkpoint inhibitors (ICIs) (Waidmann, 2018). Recently, ICIs have shown great survival benefits in patients with HCC (Chen et al., 2021). Among the numerous checkpoint molecules, programmed death ligand 1 (PD-L1) is the most studied and has the greatest clinical implications in human cancer (Wang et al., 2021). Therefore, we also evaluated the relationship between EXOSC10 and immune checkpoints. The results demonstrated that the expression of EXOSC10 was positively correlated with CD274 (PD-L1) (Fig. S3A), which indicated that EXOSC10 might increase the efficacy of PD-L1.

Tumor mutation burden (TMB) is strictly correlated with the number of neoantigens that arise in a tumor and has emerged as a potential novel biomarker in ICI therapy (Peng et al., 2021). EXOSC10 is reported to be a tumor-specific antigen in ovarian cancer (Antony et al., 2019). We performed mutation analysis to investigate EXOSC10 in HCC and found that EXOSC10 expression was significantly associated with TMB in HCC based on TCGA (Fig. S3B). Whether EXOSC10 is a neoantigen in HCC remains to be further investigated.

In vitro, we found that EXOSC10 mRNA and protein were significantly overexpressed in the three liver cancer cell lines (HepG2, MHCC97H, and Huh-7) and the knockdown of EXOSC10 inhibited the proliferation and migration of HCC cells. Furthermore, the results of the measurement of mitochondrial membrane potential, Hoechst 33342 dye, and the flow cytometer indicated that silencing of EXOSC10 can promote cell apoptosis, respectively. To explore which signaling pathway was involved in EXOSC10 inducing cell apoptosis, we found that the p53 pathway was enriched in the high expression group of EXOSC10 by GSEA and KEGG enrichment analysis. Meanwhile, considering previous studies indicated that EXOSC10 was essential for cell survival by suppressing the p53 signaling pathway (Ulmke et al., 2021). And, our findings revealed upregulation of several genes, including Bax, p21, p53, phosphorylated-p53, and E-cadherin, and downregulation of PARP, N-cadherin and Bcl-2 was observed when silencing EXOSC10. Therefore, we have shown that the elimination of EXOSC10 increases p53 phosphorylation, it may regulate the progression of HCC via the p53 pathway.

Furthermore, it is interesting that the STRING database showed that EXOSC10 may regulate the p53 signaling pathway that regulates the expression of DDX5 (DEAD box protein 5) through the PPI network (Protein–Protein Interaction) network (Fig. S4). DDX5 was important in cellular processes, which could inhibit liver tumorigenesis by stimulating autophagy (Bourgeois, Mortreux & Auboeuf, 2016; Zhang et al., 2019). However, the relationship between EXOSC10 and DDX5 deserves further study.

Conclusions

Although our research provided evidence at multiple levels to confirm that EXOSC10 is a potential prognostic biomarker and may regulate HCC progression through the p53 pathway, there were some limitations to this study. Especially, the functional analysis of EXOSC10 in HCC should be further verified in a clinical study.

Supplemental Information

Supplemental Information 1 DEGs between high- and low-EXOSC10 expression group

Click here for additional data file.

Supplemental Information 2 EXOSC10 methylation analysis and pan-cancer analysis

(A) Distribution of EXOSC10 methylation sites in various tumor types. (B) Pan-tumor analysis of EXOSC10 methylation in tumor and normal tissues. (C). The expression of EXOSC10 in different tumor types in TIMER. Note: ns: p > 0.05; *: p < = 0.05; **: p < = 0.01; ***: p < = 0.001; ****: p < = 0.0001.

Click here for additional data file.

Supplemental Information 3 The HCC-related checkpoints and the tumor mutational burden (TMB) among EXOSC10

(A–B) The association between EXOSC10 and HCC-related checkpoints. (C) The association between EXOSC10 and TMB in HCC.

Click here for additional data file.

Supplemental Information 4 PPI (Protein-Protein Interaction) network

EXOSC10 may regulate p53 signaling pathway by regulating the expression of DDX5.

Click here for additional data file.

Supplemental Information 5 DEGs between high- and low EXOSC10 expression group

2724 significant DEGs between high and low expression groups, of which 41 genes were downregulated and 2683 genes were upregulated.

Click here for additional data file.

Supplemental Information 6 Uncropped Western blots

Click here for additional data file.

Supplemental Information 7 Raw flow cytometry data

Raw data exported from the flow cytometry applied for data analyses and preparation for Fig. 7C.

Click here for additional data file.

Supplemental Information 8 Human participant data

Click here for additional data file.

Abbreviations

HCC Hepatocellular Carcinoma

EXOSC10 Exosome component 10

TIMER Tumor Immune Estimation Resource

SMART Shiny Methylation Analysis Resource Tool

TCGA The Cancer Genome Atlas

DEGs Differential expression genes

GSEA Gene Set Enrichment Analysis

OS Overall Survival

PFS Progression-free survival

CHC Chronic hepatitis C

CHB Chronic hepatitis B

GO Gene Ontology

KEGG Kyoto Encyclopedia of Genes and Genomes

ESTIMATE Estimation of Stromal and Immune cells in Malignant Tumours using Expression data

TMB Tumor mutation burden

FBS Fetal Bovine Serum

PVDF Polyvinylidene fluoride

TBST Tris–HCl solution + Tween-20

NC Negative Control

CCK-8 Cell Counting Kit-8

PBS Phosphate Buffer Saline

ICIs Immune checkpoint inhibitors

DDX5 DEAD box protein 5

PPI Protein-Protein Interaction

Additional Information and Declarations

Competing Interests

Author Contributions

Human Ethics

Data Availability

All authors declare that they have no financial, personal interests or beliefs that affect their objectivity, and have no economic or personal relationships with other people or organizations that improperly influence or prejudice their works.

Zhi-Yong Meng conceived and designed the experiments, performed the experiments, analyzed the data, prepared figures and/or tables, authored or reviewed drafts of the article, and approved the final draft.

Yu-Chun Fan conceived and designed the experiments, performed the experiments, analyzed the data, prepared figures and/or tables, authored or reviewed drafts of the article, and approved the final draft.

Chao-Sheng Zhang performed the experiments, authored or reviewed drafts of the article, and approved the final draft.

Lin-Li Zhang performed the experiments, authored or reviewed drafts of the article, and approved the final draft.

Tong Wu performed the experiments, authored or reviewed drafts of the article, and approved the final draft.

Min-Yu Nong performed the experiments, authored or reviewed drafts of the article, and approved the final draft.

Tian Wang performed the experiments, authored or reviewed drafts of the article, and approved the final draft.

Chuang Chen conceived and designed the experiments, performed the experiments, analyzed the data, prepared figures and/or tables, authored or reviewed drafts of the article, and approved the final draft.

Li-He Jiang conceived and designed the experiments, performed the experiments, analyzed the data, prepared figures and/or tables, authored or reviewed drafts of the article, and approved the final draft.

The following information was supplied relating to ethical approvals (i.e., approving body and any reference numbers):

All procedures were performed according to the Ethical Guidelines for Human Genome/Gene Research and were approved by the Ethics Committee of Affiliated Hospital of Youjiang Medical College for Nationalities (2022090501).

The following information was supplied regarding data availability:

Data are available at GitHub and Zenodo:

https://github.com/Yu-ChunFan/Zhi-Yong-Meng.git.

Yu-ChunFan. (2023). Yu-ChunFan/Zhi-Yong-Meng: V1.1.8 (v1.1.8). Zenodo. https://doi.org/10.5281/zenodo.8164736.

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
