# Peer review of "EXOSC10 is a novel hepatocellular carcinoma prognostic biomarker: a comprehensive bioinformatics analysis and experiment verification"

_PeerJ, doi:10.7717/peerj.15860_

## Round 0.1 · original submission · Major Revisions

All three reviewers have given their own opinions. Please try your best to revise and give corresponding responses.

Reviewer 1 ·

Basic reporting

The authors comprehensively evaluated the role of EXOSC10 in HCC using already published data and proof-of-concept studies using in vitro cell lines.

I mainly have minor comments on improving the way figures are presented and some missing information.

Experimental design

Figure 1A - what is the point of showing this in the main figure. There needs to be either more description in results and legends or it should go in supplementary since all it's showing is methylation sites - a simple sentence such as 10-15 methylation sites have been reported to be involved in cancer- also why did the authors looked at the methylation sites. The authors also do not explain what is island, N_shore etc.

Figure 1B- where is HCC on the plot? Not all cancers show decreased levels of methylation as reported by the authors.

Figure 1C- Again, authors mention HCC to be on the plot but i do not see it. The color legend not presented. If this is in the main more discussion on it differential expression in different tumors must be emphasized.

Figrue 2A and B sould have different x-axis labels.
Figure 2J needs scale bars
Figure 4- Showing just the top 50 differentially regulated genes does not tell much. The authors should highlight some of the genes and talk about what they mean. P-value of the GSEA should be reported for each enrichment.
Figure 8: Scale Bars missing

Validity of the findings

No Comments

Reviewer 2 ·

Basic reporting

The manuscript by Men and coworkers titled ‘EXOSC10 is a novel prognostic biomarker: A comprehensive bioinformatics analysis and experiment verification’ describes comprehensive bioinformatic analysis EXOSC10 with special emphasis on hepatocellular carcinoma. In addition, authors also provided experimental evidence to prove the oncogenic potential of EXSC10.

This study is interesting. I appreciate the authors for conducting a comprehensive analysis. Introduction and discussion were appropriately written. Methods were described adequately clear with few exceptions. Results section requires minor improvement in writing and presentation.

I ask authors to address the following minor concerns.

1) 1) Please edit the results section to make definitive statements. For correlation analysis, it is not appropriate to write that there is ‘association’. Instead, please write if it is positively or negatively correlated. While describing correlation please state if the correlation is positive or negative. In addition, please write if EXOSC10 expression increased or decreased between classes instead of writing significantly different. Please edit the relevant sentences in abstract and discussion too.
2) In methods, please elaborate on R packages used for each analysis. Authors wrote ‘R software’ very often. But it will not help the readers in reproducing the data presented here.
3) Please show higher magnification of IHC staining. In results, please describe staining pattern and localization of the protein.
4) I suggest authors to modify the nomogram and calibration curves. In nomogram, coefficient of EXOSC10 is miniscule compared to the stage suggesting that the contribution of EXOSC10 is minimal for the classifier developed. This representation of nomogram is not appropriate for visual calculation. It would be better represented as lines (for example as implemented in RMS package). Just using independent prognostic factors in multivariate analysis may improve the nomogram as well as calibration plot. The current calibration plot indicates the model is not reflecting the observed probabilities well.
5) There is no evidence in this manuscript that confirm the involvement of p53 pathway in apoptosis seen in these cells. This requires evidence showing that constitutive inactivation of p53 would prevent apoptosis observed following knockdown of EXOSC10. Therefore, please tone down the claim that EXOSC10 silencing induces apoptosis via p53 pathway.

Experimental design

No comment

Validity of the findings

No comment

Reviewer 3 ·

Basic reporting

1. Language need to be improved.
2. Resolution ratio of figures are too low. Multiple figures can not be visulized clearly.

Experimental design

1. It should be described more clearly how research fills an identified knowledge gap.
2. More details (e.g., cat No. of reagents and antibodies) should be included in the method to replicate.

Validity of the findings

1. In multiple sessions, correlations were claimed. However, whether the correlation was positive or negative was not cleared stated.
2. Line 301. The claim in the headlines of the section is contraversal to the descriptions in the folloing contents.

Additional comments

1. More clear descriptions of the figures and findings are suggested.
2. More interpretations of the results are suggested to be added.

---

## Round 0.2 · Major Revisions

Please revise the paper according to the reviewer's opinion. If the reviewer's consent is not obtained, this study will not be published.

Reviewer 1 ·

Basic reporting

The authors have addressed the issues raised by me. I recommend that it can be published now.

Experimental design

Not applicable

Validity of the findings

Not applicable

Reviewer 2 ·

Basic reporting

-

Experimental design

-

Validity of the findings

I think authors did not understand one of my comment regarding the statement related to claim that EXOSC10 induces apoptosis via p53. I pointed out that this claim is not adequately supported by the data. The data is sufficient to prove that EXOSC10 knockdown increases p53 phosphorylation. This does not establish whether this is a cause or effect. If authors want to state that EXOSC10 acts through p53, please show that EXOSC10 knock down would not induce apoptosis after homozygous deletion of TP53. Please not that this requires knock out pf TP53. Cells with p53 mutation are not suitable for this as mutant p53 is known to show gain of function activity. Therefore, mutant cell line is not equivalent to p53 knockout.

Reviewer 3 ·

Basic reporting

Suggest to further improve language.

Experimental design

no comment

Validity of the findings

The description of Fig1 is discordant in the main body and figure legend. In the main body, it is decribed as "when compared with their corresponding adjacent liver tissues (Fig1B)." However, in the Figure legend in Fig1, it is decribed as "in HCC patient samples and healthy people". Which is correct? sample from heathy peple or adjacent tissue from the same patients?

Description of Figure 6 is opposite to the conclusion in the main boday.
Figure 6 title is "Silencing EXOSC10 increases HCC cell growth and migration." However, in the main body, it is said "Silencing EXOSC10 inhibited HCC cell proliferation and migration". Which is correct?

The multiple discordances throughout the manuscript significantly decreased the liability of the concusion.

It is improdent to draw the conclusion "Silencing EXOSC10 promotes Huh-7 cells apoptosis via p53 pathway." The causility vs correlation is not sufficiently demonstrated.

---

## Round 0.3 · Minor Revisions

The author needs to revise the manuscript further. (1) All micrographs should be marked with scale. (2) The flow scatter plot on the left of Figure 7C has a problem. The compensation value is incorrect, please readjust and upload the original file. (3) All pictures should be aligned.

Reviewer 2 ·

Basic reporting

The authors addressed my concerns

Experimental design

-

Validity of the findings

-

---

## Round 0.4 · accepted · Accept

After several revisions, this manuscript basically meets the publication requirements.